# Characteristics of Carbon, Nitrogen and Phosphorus Stoichiometry and Nutrient Reabsorption in Alfalfa Leaves with Different Fall-Dormancy Levels in Northern Xinjiang, China

**Yanliang Sun** **, Xuzhe Wang, Chunhui Ma and Qianbing Zhang \***

College of Animal Science and Technology, Shihezi University, Shihezi 832003, China
\* Correspondence: qbz102@shzu.edu.cn

**Abstract:** Alfalfa productivity and cold resistance in different regions are influenced by the fall-dormancy level of alfalfa. However, it is unclear whether the stoichiometric characteristics and nutrient resorption efficiency in alfalfa leaves also vary with the fall-dormancy level. In order to further understand the differences in nutrient absorption and requirements of different fall-dormant alfalfa, we conducted field trials on 30 different fall-dormancy alfalfa cultivars for 2 consecutive years in 2020 and 2021. We investigated the concentrations of carbon, nitrogen, and phosphorus in mature and senescent alfalfa leaves; nutrient stoichiometry ratios; and the coupling relationship between nutrient reabsorption efficiency and dry matter yield. The differences in nutrient reabsorption, fall dormancy, and dry matter yield of different fall-dormant alfalfa, and the correlation between indicators were utilized to further analyze the regulatory mechanisms of nutrient reabsorption in different fall-dormancy alfalfa varieties. The results demonstrated that the nitrogen reabsorption efficiency (NRE) and phosphorus reabsorption efficiency (PRE) of leaves increased first and then decreased with the increase in fall dormancy, whereas the carbon reabsorption efficiency (CRE) showed the reverse tendency. Different fall-dormancy alfalfa varieties significantly affected the dry matter yield and nutrient absorption in the first cut, while the last cut had the lowest variable coefficient and impact. There was a significant decrease in the over-winter survival rate of alfalfa as the fall-dormancy level increased, whereas the over-summer survival rate was less affected by the fall-dormancy level. As the growth year increased, there was a significant decrease in the over-winter survival rate. Among mature leaves, the NRE and PRE showed a significant positive correlation with the C concentration, while they showed a significant negative association with the N and P concentrations. In the same cut, the dry matter yield decreased with the increase in CRE but increased with the increases in NRE and PRE, while there was no significant trend in dry matter yield and nutrient resorption efficiency (NuRE) between different cuts. Taken together, the alfalfa survival rate and dry matter yield were relatively better in the moderate fall-dormancy (fall-dormancy level, FD = 4, 5) types and fall-dormancy (FD = 3) type, with a corresponding increase in the reabsorption requirements for nitrogen and phosphorus.

**Keywords:** alfalfa; carbon; nitrogen and phosphorus stoichiometry; fall dormancy; nutrient reabsorption



## 1. Introduction

Stoichiometry provides a scientific method for exploring ecosystem nutrition, balance, and coupling correlations of main components. The research on the stoichiometric correlations among carbon, nitrogen, and phosphorus in plant leaves can reveal the laws of plant growth changes and nutrient limitation [1,2]. C is the uppermost element that makes up the dry matter in plants, and N as well as P are critical components of various proteins and genetic material [3]. There is an equilibrium correlation among all elements within the plant, meaning that plants do not regulate nutrient cycling of a single element within the plant but affect multiple elements simultaneously. For the reason that the plant

growth state and elemental stoichiometry are bound up with each other, multi-element measurements provide a better understanding of the correlations among them. Among the above measurements, C:N and C:P not only represent the ability of plants to assimilate C while absorbing nutrients but also reflect the efficiency with which plants utilize N and P, while N:P indicates the extent to which the soil environment is able to provide nutrients to plants [4,5].

On account of a reduction in temperature and light in autumn, alfalfa plants go into dormancy, which to some extent illustrates the adaptability of alfalfa varieties to changes in climate [6,7]. Alfalfa fall dormancy is a comprehensive trait that reflects both the over-wintering ability and yield potential of alfalfa varieties as well as the potential quality of alfalfa [8]. Research on alfalfa fall dormancy has fixed enormous attention on ex situ introduction experiments, physiological responses to cold tolerance, evaluation of cold resistance, frost management strategies for alfalfa, and gene research on genes as well as proteins correlated with fall dormancy. Nevertheless, rare research has been conducted on the differences between nutrient limitation and nutrient cycling within alfalfa plants. It is possible for plants to adjust the kinetic characteristics of nutrient uptake on the basis of their nutritional requirements during growth and development. A process that meets its own needs by adjusting the resorption of senescent leaves alleviates the limitation of specific nutrients by maximizing the uptake of the most limiting elements [9]. The resorption of nutrients in senescent tissues is an adaptive strategy for internal nutrient cycling during plant growth that lessens soil nutrient dependence [10]. Additionally, it is controlled by both internal gene expression and the external environment where the plant is located, and it is an adaptive change that has developed during the long-term evolution of plants [11,12]. A significant proportion of nutrients in plants can be transferred from senescent branches and leaves to other viable tissues through nutrient reabsorption, thus increasing the nutrient use efficiency of plants and enhancing the adaptation to nutrient deprivation [13]. It has been shown that the nutrient reabsorption efficiency of leaves is closely related to soil nutrient status and decreases with the increasing availability of corresponding soil nutrients. Nutrient reabsorption varies among different plant species and geographical locations, and it is susceptible to climatic conditions and soil nutrients [14]. Yuan comprehensively analyzed the nutrient reabsorption in leaves of 28 plant species in Inner Mongolia, China, and found that the nutrient reabsorption differences specifically depend on the species and life forms. Of which, the nitrogen reabsorption efficiencies ranged from 29.8% to 76.1%, with the highest nitrogen reabsorption efficiency in herbs and the lowest in legumes [15]. Van studied the effect of long-term nitrogen application on nutrient resorption in four plant species and found that there were relatively significant interspecific differences in nutrient resorption capacity. Long-term nitrogen application caused more nitrogen to be returned to the soil through litter form, and the plant became more susceptible to other nutrient limitations [14]. Lü discovered that increasing the amount of field irrigation could enhance the resorption of phosphorus from senescent alfalfa leaves, thus improving the limitation of soil phosphorus on alfalfa growth [16]. Meanwhile, a study about the effects of nitrogen and phosphorus fertilizers on the reabsorption of nitrogen, phosphorus, and potassium in alfalfa on the Loess Plateau showed that nitrogen application barely affected the nitrogen and potassium absorption in leaves but increased plant phosphorus absorption. In contrast, phosphorus application was able to increase the reabsorption of nitrogen and potassium and influenced the reabsorption of plant phosphorus in different forms [17].

Nutrient reabsorption from senescent leaves contributes to the recycling of elements, thus improving the adaptation and persistence of plants [9,18]. However, it is unclear how the changes in C, N, and P reabsorption affect the leaf stoichiometric ratios and productivity in alfalfa with various fall-dormancy levels. To better understand the nutrient cycling characteristics and adaptation mechanisms within alfalfa plants, we examined the stoichiometric endostability and nutrient reabsorption characteristics among different fall-dormancy varieties. The following hypotheses were tested in this study: (1) how the dry matter yield and survival rate of alfalfa changed with the fall-dormancy levels in

Xinjiang oasis zones; (2) how the nutrient content and stoichiometric ratio were affected by the fall-dormancy levels; and (3) how the nutrient resorption in leaves responded to the changes in leaf nutrient content and dry matter yield.

## 2. Materials and Methods

### 2.1. Experimental Site

The experiment was conducted from 2020 to 2021 at the forage-grass experimental station (44°20′ N, 86°30′ E) of Shihezi University in Shihezi City, Xinjiang. The site is located at an altitude of 420 m and has a flooded alluvial plain landscape. The average annual temperature is 5–10 °C; the accumulated temperature above 0 °C is 4020–4118 °C; the accumulated temperature above 10 °C is 3570–3729 °C; the frost-free period is 167–172 d; the sunshine hours are 2721–2818 h; the rainfall is 190–260 mm; the evaporation is 1100–1400 mm. These values are typical of the temperate continental arid climate. As a deep salinized and alkaline soil type, the soil in the test area is a typical gray desert soil (Chinese soil taxonomic classification) with a high PH, poor fertility, and a high calcium carbonate concentration. The basic physical and chemical properties of the 0–20 cm soil layer in the test field are: total nitrogen 1180 mg·kg$^{-1}$, total phosphorus 0.53 g·kg$^{-1}$, available nitrogen 33.20 mg·kg$^{-1}$, available phosphorus 30.21 mg·kg$^{-1}$, and available potassium 119.8 mg·kg$^{-1}$. The meteorological data during the test period are shown in Figure 1.

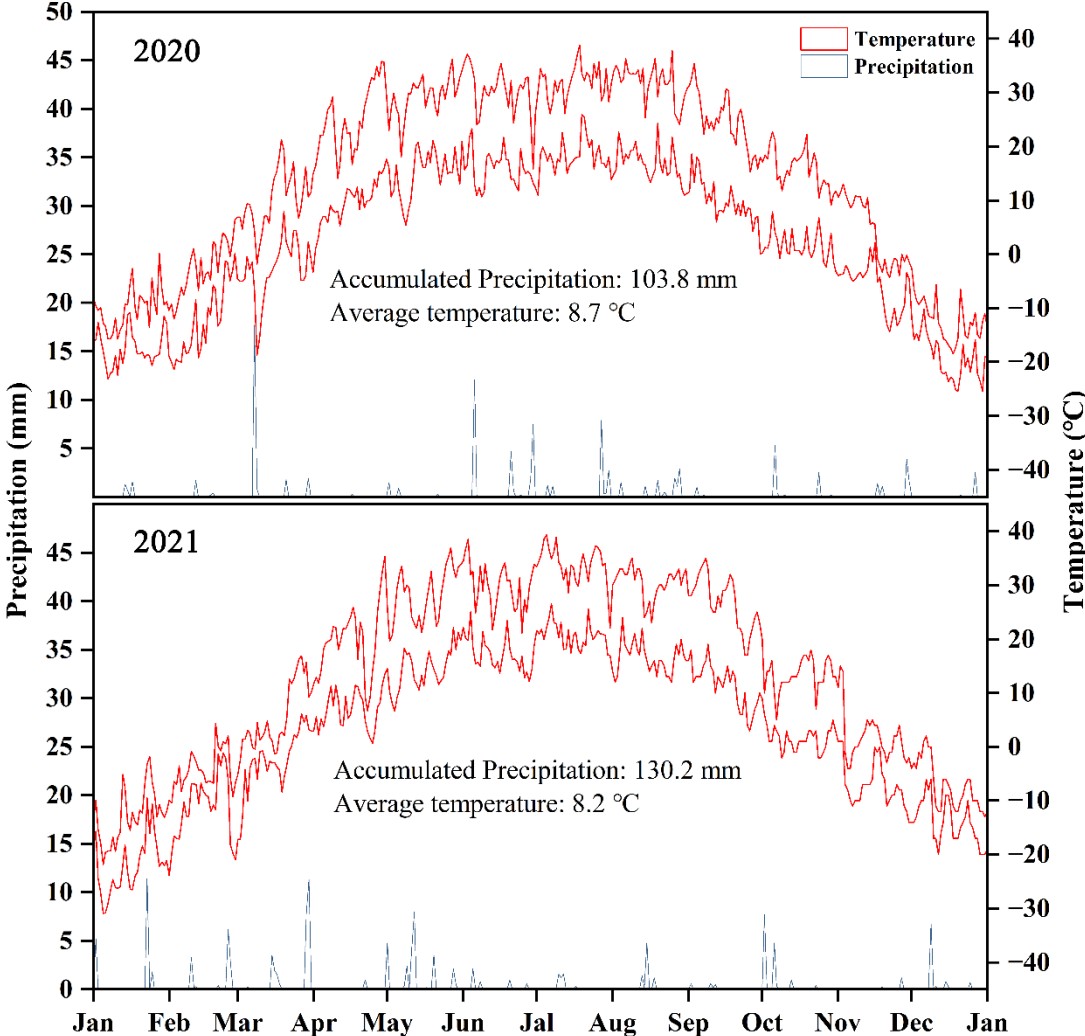

**Figure 1.** Climate data during the experimental period.

## 2.2. Experimental Materials and Experimental Design

The experiment was conducted in a single-factor randomized group design with 95 alfalfa varieties planted in the test field. For research purposes, we selected 3 common alfalfa varieties at each fall-dormancy level (1–10), a total of 30 alfalfa varieties, including 13 Chinese local alfalfa varieties and 17 foreign-introduced alfalfa varieties. The fall-dormancy levels of alfalfa were referred to previous studies [18] and provided by companies. The variety name, fall-dormancy type, source, and origin of all alfalfa materials are shown in Table 1.

**Table 1.** Cultivars and sources of alfalfa.

| Cultivar | Fall Dormancy Rate | Type Fall Dormancy | Source |
|---|---|---|---|
| Xinmu NO.1 | level 1 | Extreme fall-dormancy type | |
| Gongnong NO.1 | level 1 | Extreme fall-dormancy type | |
| Zhaodong | level 1 | Extreme fall-dormancy type | |
| Aohan | level 2 | Fall-dormancy type | |
| Xinmu NO.2 | level 2 | Fall-dormancy type | |
| Zhongmu NO.1 | level 2 | Fall-dormancy type | |
| Zhongmu NO.2 | level 3 | Fall-dormancy type | |
| Concept | level 3 | Fall-dormancy type | Institute of |
| Zhongmu NO.3 | level 3 | Fall-dormancy type | Grassland |
| Adrenalim | level 4 | Moderate fall-dormancy type | Research of CAAS |
| Xinjiang Daye | level 4 | Moderate fall-dormancy type | |
| Victoria | level 6 | Moderate fall-dormancy type | |
| Ghillie | level 6 | Moderate fall-dormancy type | |
| Yumu NO.1 | level 7 | Non-fall-dormancy type | |
| Gannong NO.5 | level 8 | Non-fall-dormancy type | |
| Liangmu NO.1 | level 8 | Non-fall-dormancy type | |
| WL325HQ | level 4 | Moderate fall-dormancy type | |
| WL363HQ | level 5 | Moderate fall-dormancy type | |
| WL366HQ | level 5 | Moderate fall-dormancy type | Beijing Zhengdao |
| WL525HQ | level 8 | Non-fall-dormancy type | Ecological |
| WL903HQ | level 9 | Extreme non-fall-dormancy type | Technology Co. |
| WL656HQ | level 9 | Extreme non-fall-dormancy type | |
| WL712HQ | level 10 | Extreme non-fall-dormancy type | |
| Paola | level 5 | Moderate fall-dormancy type | |
| Sandili | level 6 | Moderate fall-dormancy type | |
| Sardi7 | level 7 | Non-fall-dormancy type | Tianjin Bailv |
| Blue moon | level 7 | Non-fall-dormancy type | International |
| Pegasis | level 9 | Extreme non-fall-dormancy type | Grass Industry Co. |
| Sardi10 | level 10 | Extreme non-fall-dormancy type | |
| UC1887 | level 10 | Extreme non-fall-dormancy type | |

Alfalfa was planted on 27 April 2018, by manual bunch planting, and to avoid the autotoxic allelopathy caused by continuous cropping, the fields with many years of other crop rotations were selected as test plots, where the previous crop was cotton (*Gossypium* spp.). Meanwhile, wheat was selected as a protection crop in the first year for planting alfalfa in this study in order to reasonably utilize soil resources and suppress weed growth. Alfalfa was sown at a row spacing of 20 cm, a depth of 2 cm, and a seeding volume of 18 kg·ha$^{-1}$. After sowing, the seeds were suppressed to ensure the complete contact between alfalfa seeds and soil. The drip irrigation belt is shallowly buried 8–10 cm below the topsoil layer with the separation distance of 60 cm, and the drip head spacing is 20 cm. The area of the plots was 3 m × 10 m = 30 m$^2$, each alfalfa variety had three repeated plots, and a 1.5 m wide pedestrian crossing was installed between the plots to avoid water and nutrient penetration. During the experiment, the irrigation amount during the whole growth period was about 6750 m$^3$·ha$^{-1}$, and the seedling water was dripped out in time after sowing. The other drip irrigation time during the alfalfa growth period was 10 d before and 3 to 5 d after mowing of each alfalfa cut (the moist peaks between the two drip irrigation belts are connected), and 30 kg·ha$^{-1}$ of monoammonium phosphate (containing P$_2$O$_5$ 52%) was dripped with water during the branching period after alfalfa re-greening and after mowing of each alfalfa cut. Except for water and fertilizer factors, additional field

management such as weeding and pest control were managed according to the local drip irrigation fields with high-yield alfalfa.

### 2.3. Measurement Indexes and Methods

At the early flowering stage of alfalfa (5–10% flowering), three quadrats (1 m × 1 m) with constant growth vigor and representativeness in each plot were randomly selected. The alfalfa in these quadrats was weighed and recorded the fresh weight after manual mowing (leaving a stubble of 5 cm), while three fresh grass samples of about 500 g were taken from each plot and brought to the laboratory, killed at 105 °C for 30 min, then dried at 65 °C to constant weight to calculate the moisture content and to convert into the dry matter yield.

On 2 October 2019, a 2 m long sample section was randomly selected in each plot line. The number of surviving plants in the sample section was counted and recorded as the initial number of surviving plants, and repeated three times. The number of surviving alfalfa trees was counted on 8 April 2020, 4 October 2020, 8 April 2021, and 29 September 2021, and the survival rate was determined by the ratio of the subsequent surviving trees to the initial surviving trees.

The alfalfa samples were collected at the early flowering stage. Senescent leaves which were naturally senescent, withered and yellow, free of disease and insect damage, not falling off naturally, and able to fall off when the alfalfa plant was gently shaken were selected; mature leaves which were dark green, fully extended, healthy without disease or insect damage, and trifoliate at the leaf axils were selected. Sufficient leaves must be collected from each plot for C, N, and P analysis. Both the senescent and mature leaves were brought back to the laboratory for killing at 105 °C for 10 min, followed by baking at 65 °C for constant weight, crushing, and passing through a 1.0 mm sieve. Leaf organic carbon (C) was determined using the potassium dichromate volumetric method (external heating); a Kjeldahl method was used to determine leaf total nitrogen (N) after pretreatment with $H_2SO_4$-$H_2O_2$ ablation; a molybdenum blue colorimetric method was used to determine leaf total phosphorus (P) after pretreatment with ashing. The C: N, C: P, and N: P ratios were calculated according to the abovementioned concentrations of C, N, and P in mature leaves.

### 2.4. Data Analysis

Nutrient resorption efficiency [12] was calculated by the following equation.

$$NuRE = (Numature - Nusenesced)/Numature \times 100\% \qquad (1)$$

Numature represents the nutrient concentration in alfalfa mature leaves, and Nusenesced represents the nutrient concentration in senescent leaves, where C, N, and P nutrient reabsorption are expressed as CRE, NRE, and PRE, respectively. Considering the inconsistency of cuts between years, the statistical analysis of differences among the cultivars with different fall-dormancy was performed using one-way ANOVA in SPSS 22 (SPSS Inc., Chicago, IL, USA); the correlations between fall-dormancy and leaf C, N, and P were evaluated by the models $y = ax + b$ and $y = ax^2 + bx + c$; the correlations between C, N, P, and NuRE in mature leaves were evaluated by the model $y = ax + b$, while dry matter yield was presented as bubble size in bubble plots; Origin Pro 2020 (Origin Lab, Northampton, MA, USA) was used for plotting.

## 3. Results

### 3.1. C, N, and P Concentrations in Mature Leaves of Alfalfa with Different Fall-Dormancy Levels

The concentrations of C, N, and P in mature alfalfa leaves varied regularly among different fall-dormancy levels (Figure 2, Table 2), with the concentrations of C, N, and P ranging from 349.19 to 415.60 mg·g$^{-1}$, 34.14 to 41.15 mg·g$^{-1}$, and 2.37 to 2.98 mg·g$^{-1}$, respectively. As alfalfa's fall-dormancy level increased, the C concentration in mature leaves showed a trend of increasing first and then decreasing, and it gradually decreased with the increase in cut times. In the first two cuts of 2020 and the first cut of 2021, the C

concentration in mature alfalfa leaves with fall-dormancy level 4 was significantly higher than that of alfalfa with fall-dormancy levels 9 and 10 ($p < 0.05$), while the differences in the C concentration in mature alfalfa leaves with each fall-dormancy level in the third cut of 2020 and the second cut of 2021 were not significant ($p > 0.05$). N and P concentrations in mature alfalfa leaves tended to decrease first and then increase with the increasing fall-dormancy levels. N concentrations in mature alfalfa leaves with fall-dormancy levels 9 and 10 were significantly higher than those with fall-dormancy levels 3 and 4 in the first cut of 2020 and both cuts of 2021 ($p < 0.05$). P concentrations in mature alfalfa leaves with fall-dormancy levels 9 and 10 were significantly ($p < 0.05$) higher than those of alfalfa with fall-dormancy levels 3, 4, and 5 in the first two cuts of 2020 and 2021.

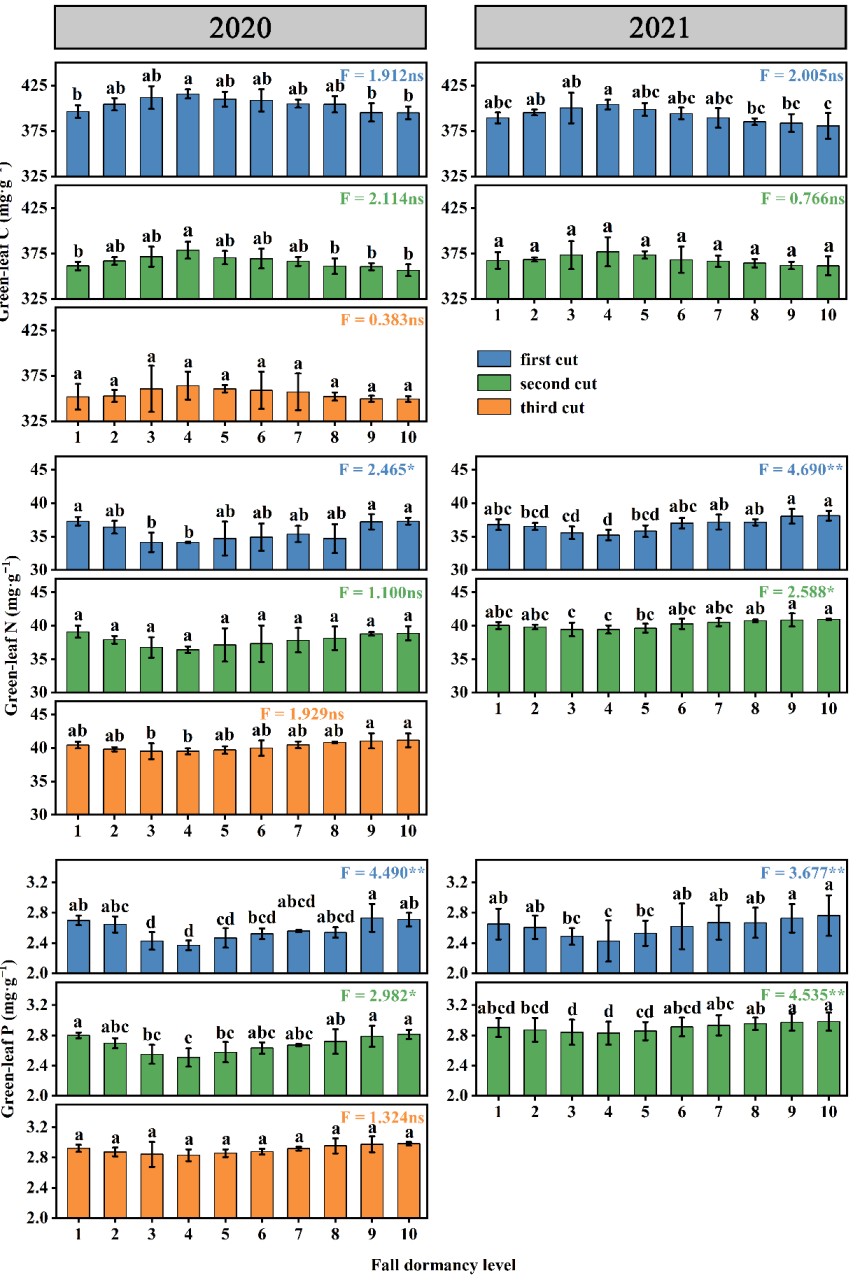

**Figure 2.** Stoichiometric characteristics of mature alfalfa leaves with different fall-dormancy levels (2020–2021). Different small letters indicate significant differences between different fall dormancy levels ($p < 0.05$). ns indicates no significant difference between different fall dormancy levels ($p > 0.05$), * indicates significant difference between different fall dormancy levels ($p < 0.05$) and ** indicates extremely significant difference between different fall dormancy levels ($p < 0.01$).

**Table 2.** Correlation analysis between fall-dormancy levels and C, N, and P concentrations in mature alfalfa leaves (2020–2021).

| Index | Year | Cut | CV (%) | Linear Model | | | | Nonlinear Model | | | |
| | | | | Y = ax + b | $R^2$ | F | P | Y = ax² + bx + c | $R^2$ | F | P |
|---|---|---|---|---|---|---|---|---|---|---|---|
| Green-leaf C | 2020 | 1 | 1.78 | y = −0.884x + 409.460 | 0.139 | 1.286 | 0.290 | y = −0.777x² + 7.668x + 392.360 | 0.8258 | 16.586 | 0.002 |
| | | 2 | 1.79 | y = −1.070x + 372.150 | 0.244 | 2.574 | 0.147 | y = −0.624x² + 5.795x + 358.430 | 0.774 | 11.991 | 0.006 |
| | | 3 | 1.48 | y = −0.664x + 359.300 | 0.146 | 1.368 | 0.276 | y = −0.548x² + 5.361x + 347.250 | 0.783 | 12.593 | 0.005 |
| | 2021 | 1 | 1.97 | y = −1.726x + 401.510 | 0.459 | 6.792 | 0.031 | y = −0.603x² + 4.907x + 388.250 | 0.818 | 15.745 | 0.003 |
| | | 2 | 1.37 | y = −1.091x + 374.190 | 0.426 | 5.949 | 0.041 | y = −0.378x² + 3.062x + 365.880 | 0.753 | 10.693 | 0.007 |
| Green-leaf N | 2020 | 1 | 3.67 | y = 0.076x + 35.205 | 0.031 | 0.254 | 0.628 | y = 0.148x² − 1.548x + 38.452 | 0.778 | 12.239 | 0.005 |
| | | 2 | 2.42 | y = 0.093x + 37.320 | 0.094 | 0.826 | 0.390 | y = 0.098x² − 0.981x + 39.467 | 0.758 | 10.921 | 0.007 |
| | | 3 | 1.56 | y = 0.152x + 39.411 | 0.533 | 9.141 | 0.017 | y = 0.047x² − 0.360x + 40.434 | 0.855 | 20.591 | <0.001 |
| | 2021 | 1 | 2.66 | y = 0.227x + 35.496 | 0.495 | 7.808 | 0.023 | y = 0.066x² − 0.502x + 36.954 | 0.765 | 11.387 | 0.006 |
| | | 2 | 1.45 | y = 0.158x + 39.315 | 0.676 | 16.789 | 0.003 | y = 0.028x² − 0.154x + 39.939 | 0.816 | 15.627 | 0.003 |
| Green-leaf P | 2020 | 1 | 4.90 | y = 0.011x + 2.504 | 0.076 | 0.611 | 0.457 | y = 0.013x² − 0.134x + 2.796 | 0.727 | 9.296 | 0.011 |
| | | 2 | 4.02 | y = 0.008x + 2.608 | 0.025 | 1.219 | 0.302 | y = 0.015x² − 0.159x + 2.943 | 0.539 | 12.761 | 0.005 |
| | | 3 | 1.90 | y = 0.013x + 2.832 | 0.493 | 7.766 | 0.034 | y = 0.004x² − 0.035x + 2.927 | 0.853 | 20.335 | <0.001 |
| | 2021 | 1 | 4.04 | y = 0.022x + 2.496 | 0.380 | 5.008 | 0.056 | y = 0.008x² − 0.066x + 2.671 | 0.713 | 8.879 | 0.012 |
| | | 2 | 1.88 | y = 0.014x + 2.827 | 0.616 | 13.986 | 0.006 | y = 0.003x² − 0.021x + 2.898 | 0.820 | 17.086 | 0.002 |

The equations $y = ax + b$ and $y = ax^2 + bx + c$ were used to fit the relationship between the fall-dormancy levels and C, N, and P in mature leaves. The measured results for 2 consecutive years (Table 2) showed that the $R^2$ values between the fall-dormancy levels and C, N, and P concentrations in mature alfalfa leaves in the linear model were 0.139–0.459, 0.031–0.676, and 0.025–0.616, respectively, and in the nonlinear model were 0.753–0.826, 0.765–0.855, and 0.539–0.853, respectively. The coefficients of variation (CV) of C, N, and P concentrations in mature alfalfa leaves of the first cut were significantly greater than those of other cuts, except for the C concentration in mature alfalfa leaves in 2020.

### 3.2. Stoichiometry Ratios of Elements in Mature Alfalfa Leaves with Different Fall-Dormancy Levels

The stoichiometry ratios of elements in mature alfalfa leaves with different fall-dormancy levels are shown in Figures 3 and 4. C: N, C: P, and N: P of alfalfa in 2020 and 2021 showed a trend of increasing first and then decreasing with the increase in fall-dormancy levels, where C: N of alfalfa with different fall-dormancy levels in the first cut was significant ($p < 0.05$); C: P of the first two cuts was significant ($p < 0.05$) among all treatments; and N: P of alfalfa with different fall-dormancy levels was not significant among all cuts ($p > 0.05$).

### 3.3. C, N, and P Concentrations in Senescent Alfalfa Leaves with Different Fall-Dormancy Levels

The C, N, and P concentrations in senescent alfalfa leaves likewise showed some regularity among different fall-dormancy levels (Figure 5, Table 3), with the C, N, and P concentrations ranging from 349.30 to 395.56 mg·g$^{-1}$, 16.67 to 23.75 mg·g$^{-1}$, and 0.98 to 1.98 mg·g$^{-1}$, respectively. The C concentration in senescent alfalfa leaves tended to increase first and then decrease with the increase in fall-dormancy level, but the difference was not significant ($p > 0.05$) among all alfalfa varieties at each fall-dormancy level. Both N and P concentrations tended to decrease first and then increase with the increasing fall-dormancy levels, where the N concentration in senescent alfalfa leaves with fall-dormancy level 10 was significantly greater than that of alfalfa with fall-dormancy level 4 in both 2020 and 2021 ($p < 0.05$). The P concentration in senescent leaves differed significantly ($p < 0.05$) in the first cut in 2020, and alfalfa with fall-dormancy level 9 was significantly higher ($p < 0.05$) than alfalfa with fall-dormancy levels 3, 4, and 5, while the P concentration in senescent leaves in all other cuts was not significant ($p > 0.05$).

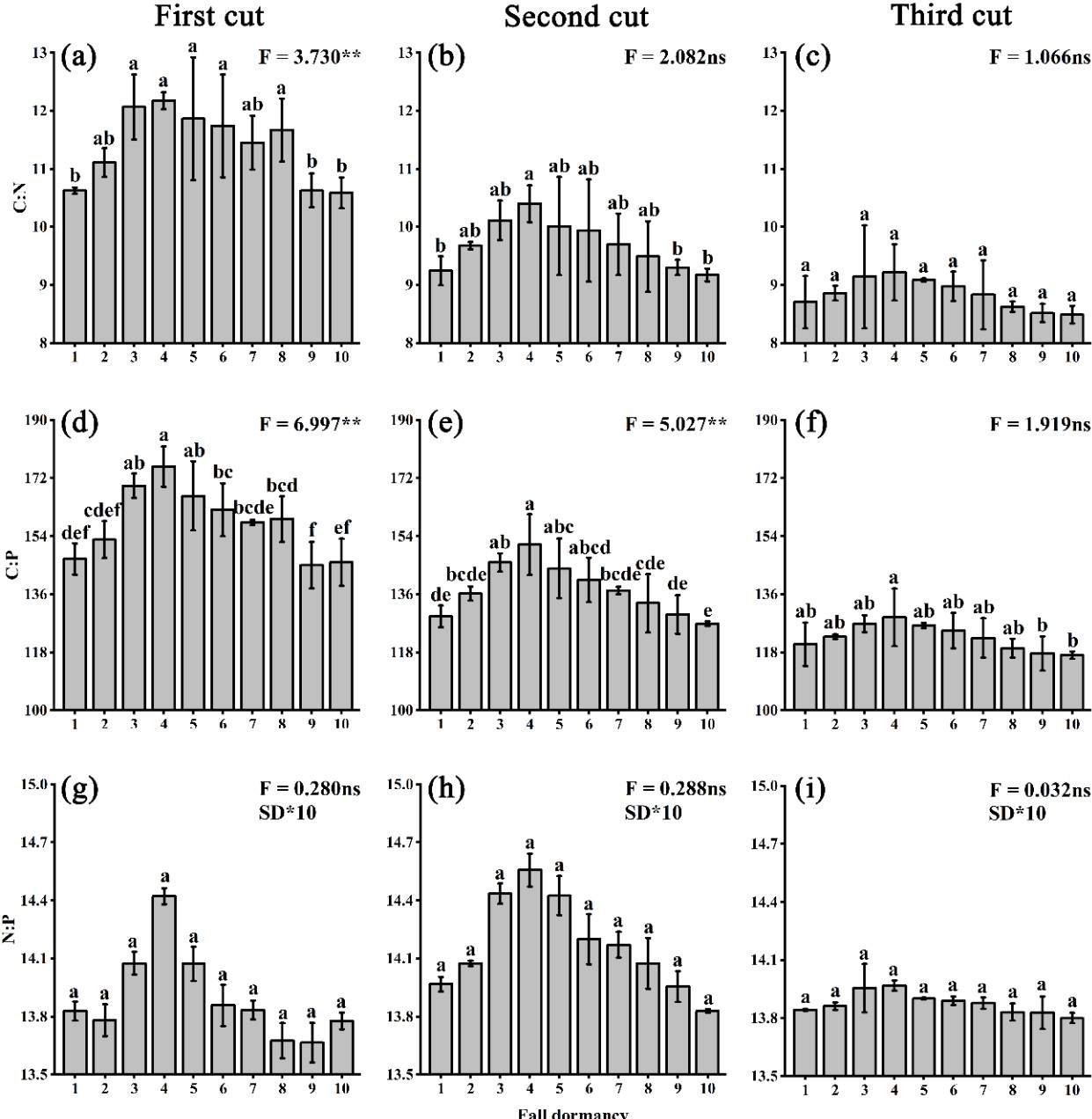

**Figure 3.** Stoichiometric ratios of elements in mature alfalfa leaves with different fall-dormancy levels in 2020. (**a,d,g**) are the first cut; (**b,e,h**) are the second cut; (**c,f,i**) are the third cut. SD*10 means that the standard error should be multiplied by 10. Different small letters indicate significant differences between different fall dormancy levels ($p < 0.05$). ns indicates no significant difference between different fall dormancy levels ($p > 0.05$), * indicates significant difference between different fall dormancy levels ($p < 0.05$) and ** indicates extremely significant difference between different fall dormancy levels ($p < 0.01$).

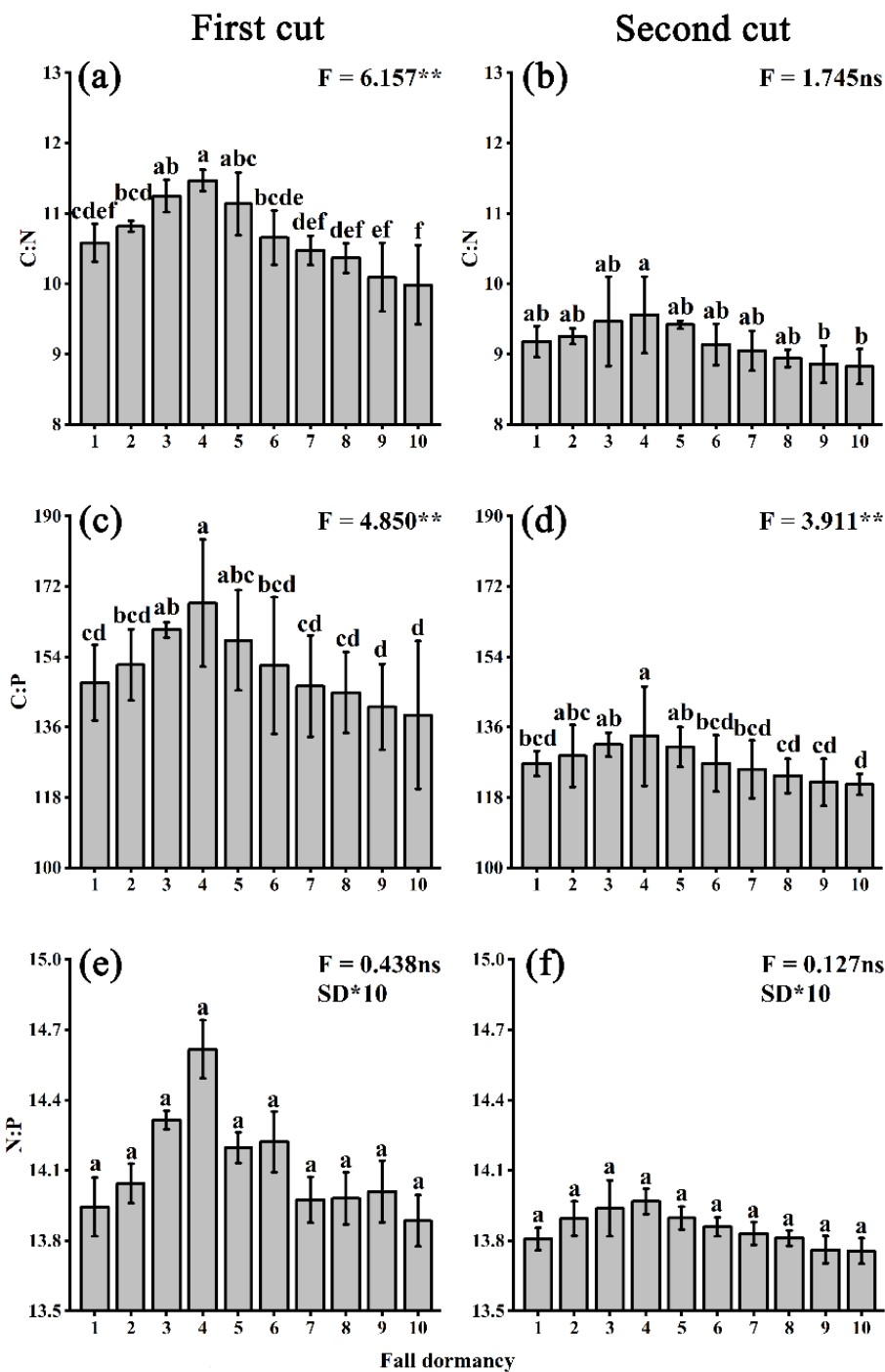

**Figure 4.** Stoichiometric ratios of elements in mature alfalfa leaves with different fall-dormancy levels in 2021. (**a,c,e**) are the first cut; (**b,d,f**) are the second cut. SD*10 means that the standard error should be multiplied by 10. Different small letters indicate significant differences between diferent fall dormancy levels ($p < 0.05$). ns indicates no significant difference between different fall dormancy levels ($p > 0.05$), * indicates significant difference between different fall dormancy levels ($p < 0.05$) and ** indicates extremely significant difference between different fall dormancy levels ($p < 0.01$).

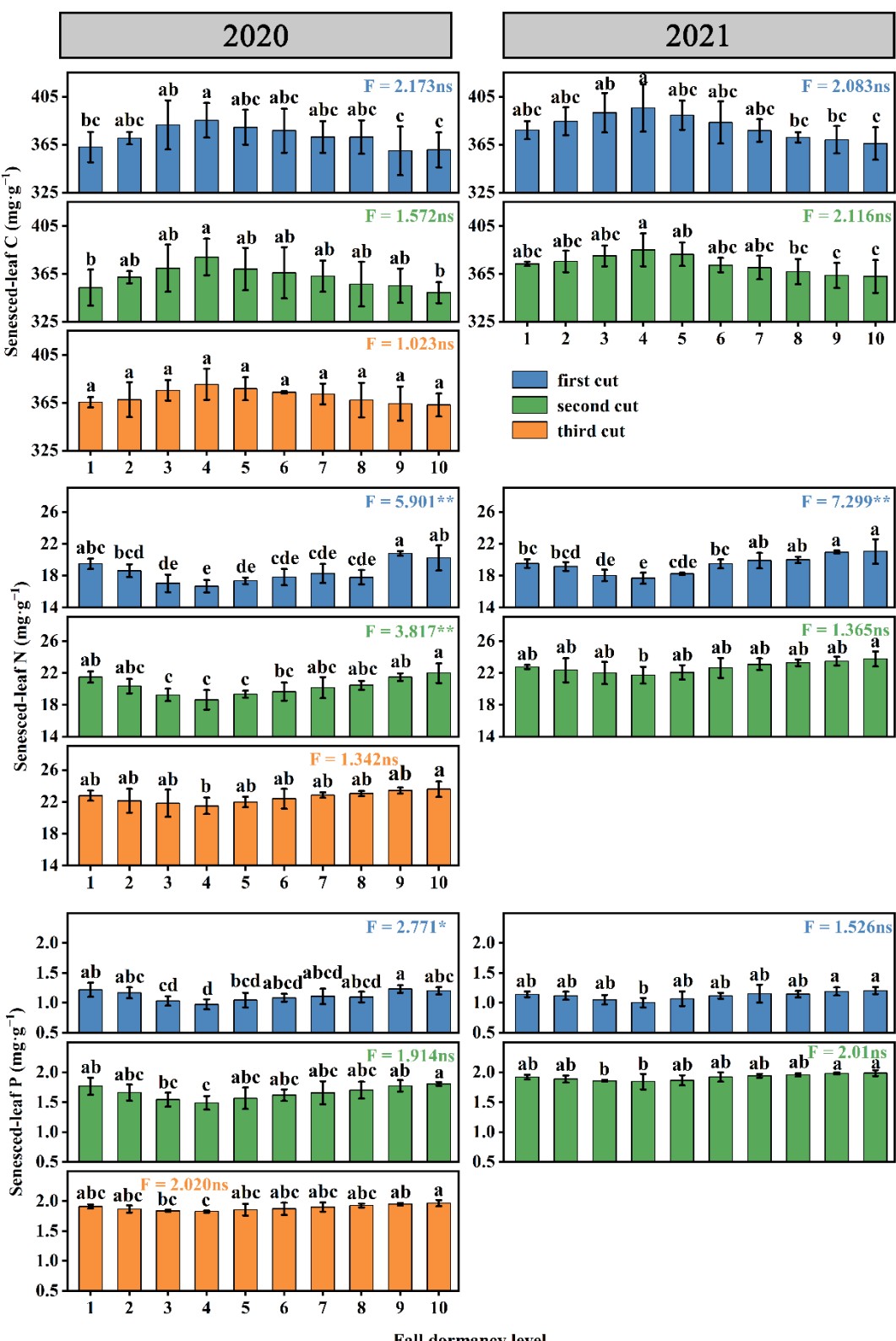

**Figure 5.** Stoichiometric characteristics of senescent alfalfa leaves with different fall-dormancy levels. Different small letters indicate significant differences between different fall dormancy levels ($p < 0.05$). ns indicates no significant difference between different fall dormancy levels ($p > 0.05$), * indicates significant difference between different fall dormancy levels ($p < 0.05$) and ** indicates extremely significant difference between different fall dormancy levels ($p < 0.01$).

**Table 3.** Correlation analysis between fall-dormancy levels and C, N, and P concentrations in senescent alfalfa leaves (2020–2021).

| Index | Year | Cut | CV (%) | Linear Model | | | | Non-Linear Model | | | |
|---|---|---|---|---|---|---|---|---|---|---|---|
| | | | | $y = ax + b$ | $R^2$ | $F$ | $P$ | $y = ax^2 + bx + c$ | $R^2$ | $F$ | $P$ |
| Senesced- leaf C | 2020 | 1 | 2.39 | $y = -1.160x + 378.290$ | 0.157 | 1.487 | 0.257 | $y = -0.930x^2 + 9.065x + 357.840$ | 0.801 | 14.047 | 0.004 |
| | | 2 | 2.48 | $y = -1.241x + 369.160$ | 0.175 | 1.694 | 0.229 | $y = -0.914x^2 + 8.807x + 349.060$ | 0.781 | 12.490 | 0.005 |
| | | 3 | 1.59 | $y = -0.676x + 374.300$ | 0.121 | 1.103 | 0.324 | $y = -0.634x^2 + 6.2947x + 360.360$ | 0.802 | 14.197 | 0.003 |
| | 2021 | 1 | 2.65 | $y = -2.257x + 392.850$ | 0.461 | 6.840 | 0.031 | $y = -0.782x^2 + 6.3417x + 375.650$ | 0.815 | 15.402 | 0.003 |
| | | 2 | 2.00 | $y = -1.784x + 382.920$ | 0.521 | 8.718 | 0.018 | $y = -0.499x^2 + 3.704x + 371.950$ | 0.782 | 12.595 | 0.005 |
| Senesced- leaf N | 2020 | 1 | 7.43 | $y = 0.187x + 17.375$ | 0.171 | 1.661 | 0.233 | $y = 0.139x^2 - 1.341x + 20.431$ | 0.776 | 12.132 | 0.005 |
| | | 2 | 5.43 | $y = 0.141x + 19.509$ | 0.149 | 1.406 | 0.270 | $y = 0.123x^2 - 1.212x + 22.214$ | 0.880 | 25.511 | <0.001 |
| | | 3 | 3.08 | $y = 0.162x + 21.651$ | 0.499 | 7.957 | 0.023 | $y = 0.052x^2 - 0.413x + 22.802$ | 0.831 | 17.261 | 0.002 |
| | 2021 | 1 | 5.95 | $y = 0.266x + 17.933$ | 0.485 | 7.540 | 0.025 | $y = 0.084x^2 - 0.658x + 19.780$ | 0.796 | 13.632 | 0.004 |
| | | 2 | 3.01 | $y = 0.170x + 21.772$ | 0.570 | 10.533 | 0.012 | $y = 0.046x^2 - 0.339x + 22.790$ | 0.839 | 18.181 | 0.002 |
| Senesced- leaf P | 2020 | 1 | 7.72 | $y = 0.007x + 1.080$ | 0.052 | 0.442 | 0.525 | $y = 0.009x^2 - 0.094x + 1.280$ | 0.706 | 8.401 | 0.014 |
| | | 2 | 6.40 | $y = 0.015x + 1.575$ | 0.182 | 1.780 | 0.219 | $y = 0.011x^2 - 0.105x + 1.816$ | 0.807 | 14.596 | 0.003 |
| | | 3 | 2.46 | $y = 0.011x + 1.833$ | 0.467 | 6.996 | 0.030 | $y = 0.004x^2 - 0.032x + 1.919$ | 0.876 | 24.819 | <0.001 |
| | 2021 | 1 | 5.57 | $y = 0.012x + 1.052$ | 0.360 | 4.505 | 0.067 | $y = 0.005x^2 - 0.041x + 1.158$ | 0.712 | 8.646 | 0.013 |
| | | 2 | 2.67 | $y = 0.013x + 1.845$ | 0.570 | 10.605 | 0.012 | $y = 0.003x^2 - 0.023x + 1.916$ | 0.804 | 14.347 | 0.003 |

To determine the $R^2$ values between the fall-dormancy levels of alfalfa and the C, N, and P concentrations of senescent leaves (Table 3), the $y = ax + b$ model was adopted and the $R^2$ values were 0.121–0.521, 0.171–0.521, and 0.052–0.570, respectively. Meanwhile, in the $y = ax^2 + bx + c$ model, the $R^2$ values were 0.781–0.815, 0.776–0.880, and 0.706–0.876, respectively. Among them, the coefficients of variation of C, N and P concentrations in senescent leaves of the first cut were greater than those of other cuts.

### 3.4. Nutrient Resorption Characteristics of Alfalfa with Different Fall-Dormancy Levels

All alfalfa leaf CRE tended to decrease first and then increase with the increasing fall-dormancy levels, while both leaf NRE and PRE tended to increase first and then decrease (Figure 6). The CRE of alfalfa leaves with fall-dormancy level 9 was significantly higher ($p < 0.05$) than that of alfalfa with fall-dormancy levels 3, 4, 5, and 6 in the first cut of 2020, while the differences were not significant ($p > 0.05$) in the other cuts. The NRE of alfalfa leaves with fall-dormancy level 4 was significantly higher than that of alfalfa with fall-dormancy level 10 ($p < 0.05$), with significant differences between the first two cuts of 2020 and the first cut of 2021 ($p < 0.05$) and non-significant differences between the other two cuts ($p > 0.05$). The PRE of alfalfa leaves with fall-dormancy levels 9 and 10 was significantly lower than that of other treatments in the first two cuts of 2020 and 2021 ($p < 0.05$), while there was no significant difference between the treatments in the third cut of 2020 ($p > 0.05$).

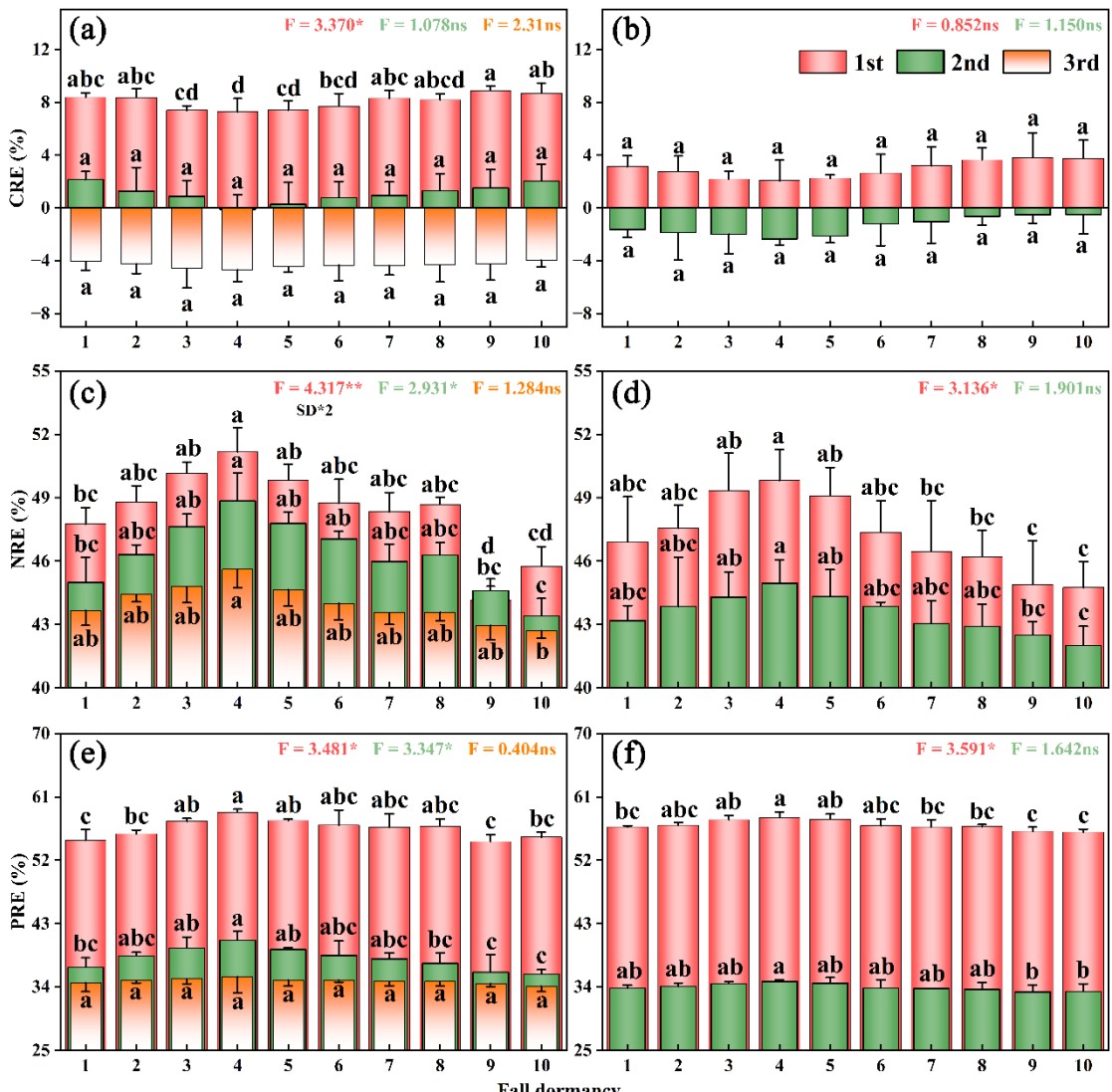

**Figure 6.** Nutrient reabsorption efficiency of alfalfa with different fall-dormancy levels, where (**a**,**c**,**e**) are 2020 test data, and (**b**,**d**,**f**) are 2021 test data. Different small letters indicate significant differences between different fall dormancy levels ($p < 0.05$). ns indicates no significant difference between different fall dormancy levels ($p > 0.05$), * indicates significant difference between different fall dormancy levels ($p < 0.05$) and ** indicates extremely significant difference between different fall dormancy levels ($p < 0.01$).

*3.5. Comparison of Survival Rates of Alfalfa Varieties with Different Fall-Dormancy Levels*

Statistics on the survival rate of alfalfa from 2020 to 2021 (Figure 7) showed that the over-winter survival rate was significantly lower than the over-summer survival rate; the decreasing trend of over-winter survival rate in 2020 was significantly higher than that in 2021; and the survival rate of alfalfa showed a decreasing trend as the fall-dormancy level increased.

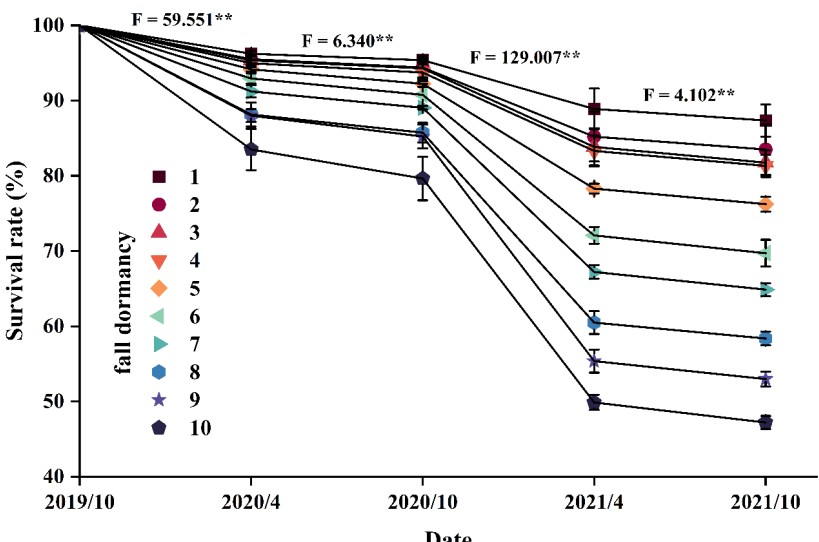

**Figure 7.** Comparison of survival rates of alfalfa with different fall-dormancy levels. *F*-values and *p*-values are ANOVA results for individual stage survival rates. ** indicates extremely significant difference between different fall dormancy levels ($p < 0.01$).

### 3.6. Comparison of Dry Matter Yield of Alfalfa with Different Fall-Dormancy Levels

The dry matter yield of alfalfa with different fall-dormancy levels is shown in Figure 8. With the increase in fall-dormancy level, the dry matter yield of alfalfa showed a trend of increase first and then decrease. In 2020, the highest cumulative annual dry matter yield was 20.33 t·ha$^{-1}$ in the alfalfa with fall-dormancy level 4, and the lowest was 13.28 t·ha$^{-1}$ in the alfalfa with fall-dormancy level 9. In 2021, the highest cumulative annual dry matter yield was 12.04 t·ha$^{-1}$ in the alfalfa with fall-dormancy level 4, and the lowest was 6.59 t· ha$^{-1}$ in the alfalfa with fall-dormancy level 10.

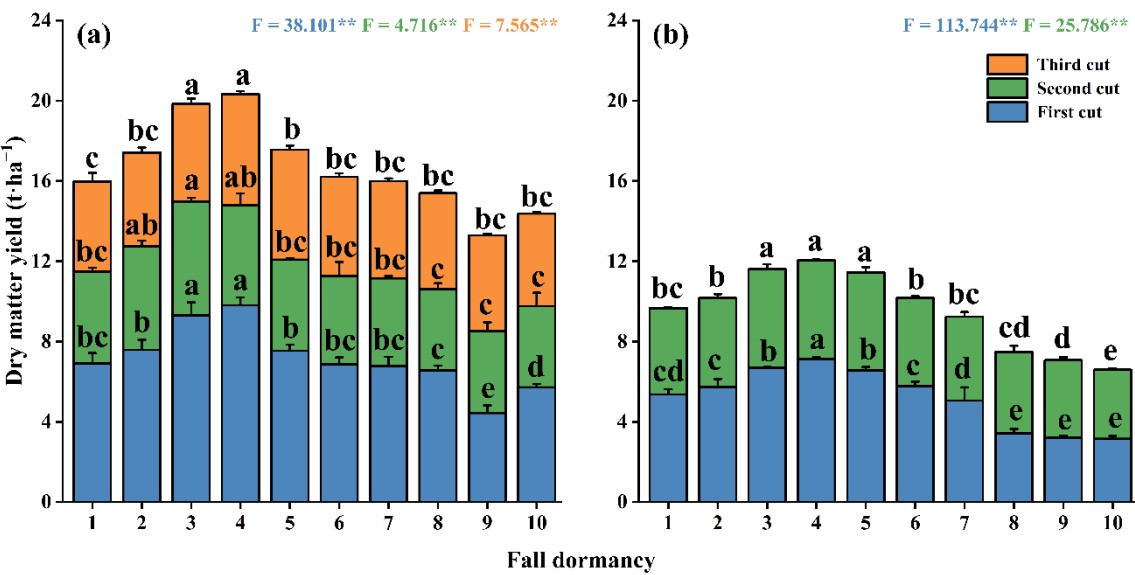

**Figure 8.** Annual cumulative dry matter yield of alfalfa with different fall-dormancy levels, where (**a**) is 2020 test data, and (**b**) is 2021 test data. Different small letters indicate significant differences between different fall dormancy levels ($p < 0.05$). ** indicates extremely significant difference between different fall dormancy levels ($p < 0.01$).

### 3.7. Relationship among C, N, and P, Elemental Reabsorption, and Dry Matter Yield in Mature Leaves

There was a significant correlation between nutrient resorption efficiency and C, N, and P concentrations in mature leaves (Figure 9) ($p < 0.05$). Among them, CRE showed a significant positive correlation ($p < 0.05$) with N and P, while a significant negative correlation ($p < 0.05$) with C concentration was shown in mature leaves. NRE and PRE showed a significant positive correlation ($p < 0.05$) with C concentration, while a significant negative correlation ($p < 0.05$) with N and P concentration was shown in mature leaves. Meanwhile, with the increase in C concentration, NRE, and PRE and the decrease in N and P concentration and CRE in mature leaves, the dry matter yield of alfalfa showed a definite increasing trend.

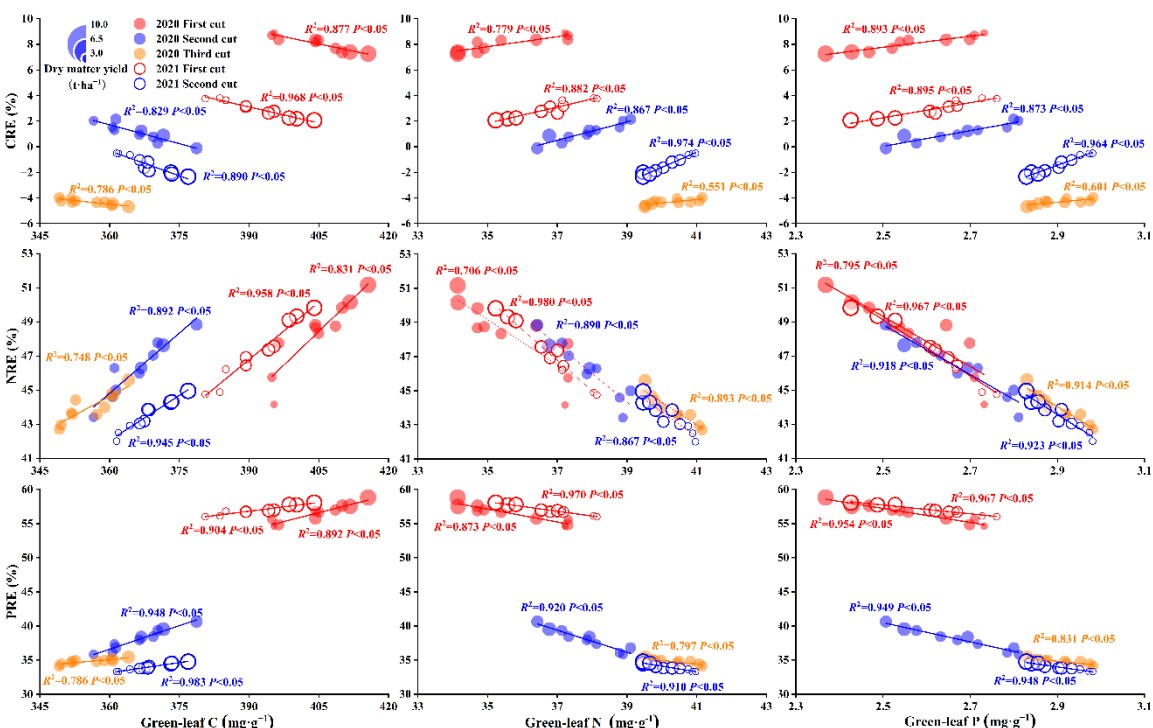

**Figure 9.** Relationship among C, N, and P, element reabsorption, and dry matter yield in mature leaves.

## 4. Discussion

### 4.1. Effect of Fall-Dormancy Level on Stoichiometric Characteristics of Alfalfa

C, N, and P are essential nutrients for plant growth and development, and their stoichiometric characteristics can reflect the intra-plant stoichiometric endostability and elemental interrelationships [3]. This study showed that alfalfa fall-dormancy levels could significantly affect leaf C, N, and P concentrations, with the medium fall-dormancy type (level 4) alfalfa obtaining the highest C and the lowest N and P concentrations. Based on previous studies, we think the concentrations of C, N, and P in different fall-dormancy alfalfa are inconsistent because of the following two aspects. On the one hand, the response of different fall-dormant alfalfa to the external environment is inconsistent, with the most significant effect of soil temperature and light duration, and the effect of light is about 75% of the soil temperature. The effect of soil temperature in the early growth period of medium fall-dormancy alfalfa is close to that of fall-dormancy alfalfa, while the effect of light duration is only half of that of fall-dormancy alfalfa varieties. However, as time goes on, the effect of light on moderate fall-dormancy alfalfa varieties decreased significantly, while the non-fall-dormancy alfalfa varieties were influenced only by soil temperature and no longer by light. This has an effect on the whole-plant biomass, photosynthetic carbon fixation, root nutrient uptake, and persistence after alfalfa re-greening [19]. On the other hand, because the low fall-dormancy alfalfa enters a dormant state very early in

the previous fall, it consumes less nutrients in the lower ground root system and stores a large amount of carbohydrates and proteins for alfalfa growth after over-wintering and re-greening [20]. In this study, we also found that alfalfa dry matter yield was highest at fall-dormancy levels 3, 4, and 5, and that the higher yield increased nutrient demand, leading to a decrease in soil nutrient content, which reduced the nutrient cycling and metabolic activity, ultimately leading to a reduction in the nutrient feedback mechanism between alfalfa and soil. While higher carbon fixation also diluted the N and P content in plants, the low biomass of alfalfa with high and low fall-dormancy levels might cause a nutrient "concentration" effect [12].

The interrelationship between the major nutrients and their stoichiometric ratios in plants is a combination of the plant's nutrient allocation strategy and adaptation to the external environment. A high C: N and C: P ratios indicate that plants use N and P more efficiently, which is a strategy for adapting to soil nutrient depletion during growth [21]. In this study, medium fall-dormancy alfalfa had higher carbon-to-nitrogen and carbon-to-phosphorus ratios, mainly due to the soil depletion caused by the high yields for three to four consecutive years. Nitrogen/phosphorus ratio can reflect the dynamic balance between soil nutrient content and plant nutrient demand to a certain extent. It is possible to identify the main limiting nutrients for alfalfa growth by analyzing its nitrogen/phosphorus concentration and nitrogen/phosphorus ratio. Koerselman [22] suggested that alfalfa growth is mainly limited by nitrogen when the nitrogen/phosphorus ratio is less than 14, and when the nitrogen/phosphorus ratio is greater than 16, the alfalfa growth is mainly limited by phosphorus. Alfalfa growth was limited by a combination of nitrogen and phosphorus when the nitrogen/phosphorus ratio was between 14 and 16. In this study, for the most of the fall-dormancy and non-fall-dormancy alfalfa types, the nitrogen/phosphorus ratios were less than 14, and they were mainly limited by N. Most of the medium fall-dormancy types were between 14 and 16 and were limited by a combination of N and P. This may be mainly due to the fact that the medium fall-dormancy alfalfa in the Shihezi region is more suitable for growth and requires more soil nutrients to be consumed during growth, so this type is more nutrient-limited. However, some studies suggest that alfalfa growth is more phosphorus-deficient. On the one hand, in this study, extra phosphorus fertilizer was added, resulting in sufficient available phosphorus for the normal growth of alfalfa. On the other hand, alfalfa, as a legume, can use rhizobia to convert the nitrogen in air into available nitrogen. Some fertilization studies have shown that moderate additions of nitrogen fertilizer can significantly increase alfalfa yield [23], indicating to some extent that the nitrogen fixed by alfalfa rhizobia is not sufficient to meet the growth requirements of alfalfa. In addition, rhizobia in soil have limitations in fixing atmospheric N. The higher yield of moderate fall-dormancy alfalfa does not match the corresponding increase in N fixation by rhizobia. Not only that, at higher soil water holding capacity, the N: P in alfalfa leaf also tends to decrease gradually with increasing growth time [24].

### 4.2. Effect of Fall-Dormancy Level of Alfalfa on Nutrient Reabsorption Characteristics of Leaves

The nutrient resorption efficiency of plant leaf is influenced by its own genetic factors and varies significantly in different functional groups [24]. The nutrient reabsorption efficiency of various fall-dormancy alfalfa types varies among cuts in this investigation, with CRE being significant only in the first cut in 2020, while NRE and PRE were not significant only in the last cut. The correlation between dry matter yield and nutrient absorption in alfalfa is strongly consistent, with yield variations determining alfalfa nutrient requirements [9]. The F-values show that the first cut has the greatest impact of alfalfa fall-dormancy level on dry matter yield in both years, which indirectly indicates that the first cut's nutritional requirements are significantly different for various fall-dormancy alfalfa types in the first cut (Figure 8). In this study, the nutrient reabsorption efficiency of medium fall-dormancy alfalfa was greater than that of fall-dormancy and non-fall-dormancy alfalfa, which also indicates that the effectiveness of soil nutrients may not be sufficient for alfalfa

growth and that the medium fall-dormancy alfalfa has a greater nutrient demand and requires more nutrient recovery to ensure its normal growth requirements.

Resorption prolongs the retention time of nutrients in alfalfa plant and provides the material basis for biomass production [25]. When the cut changed, we discovered that the resorption efficiency among various fall-dormancy alfalfa varieties tended to become smaller and smaller, which is consistent with the decrease in alfalfa yield as the cut changed. It is generally believed that alfalfa yields are highest during the first cut, and that as summer approaches, alfalfa suffers a "summer decline" in which growth slows and yields decrease. Plant leaves are sensitive to photoinhibition in summer due to the high temperature, which can directly denaturize plant proteins and liquefy membrane lipids [26]. On the one hand, high temperature and light cause rapid water loss and stomatal closure in leaves, preventing photosynthetic gas exchange and reducing carbohydrate production and photosynthesis [27]. On the other hand, high temperature affects protein activity, deactivates photosynthetic and respiratory enzyme systems and disrupts normal plant metabolism [28]. Alfalfa yield and nutrient content changed during this process, resulting in significant differences in C, N, and P concentrations between cuts.

### 4.3. Effect of Fall-Dormancy Level on Persistence and Yield of Alfalfa

Alfalfa dry matter yield and persistence is one of the key factors in evaluating the merit of alfalfa varieties and is also a major indicator of the magnitude of alfalfa productivity [29]. In this study, alfalfa varieties with fall-dormancy level 4 had the highest dry matter yield, followed by alfalfa varieties with fall-dormancy levels 3 and 5, while alfalfa varieties with fall-dormancy levels 9 and 10 had the lowest. Alfalfa varieties have different adaptability to the environment, and the latitude of the test site reflects the aspects of local climate, such as annual temperature, light duration, and snowfall. Generally, alfalfa varieties with low fall-dormancy levels at high latitudes have better production performance and persistence, and they can ensure certain economic benefits. At low latitudes, high fall-dormancy alfalfa varieties grow faster and are not affected by low temperatures and insufficient light, resulting in higher hay yields [6,30]. It has also been shown that low-latitude areas can lead to lower over-summering rates of alfalfa due to high temperatures and continuous rainfall. Additionally, soil nutrients and beneficial microorganisms may vary increasingly with the growing year due to the inconsistent growth characteristics and nutrient requirements of different alfalfa varieties [31]. In conclusion, the growth differences of different fall-dormancy alfalfa are mainly influenced by various factors such as latitude, climatic environment, and their own genetics. These factors must be taken into consideration when introducing alfalfa to exotic locations [32,33].

During the winter growth of alfalfa, its over-wintering performance is influenced by its fall-dormancy, and the over-wintering rate is an important indicator of the persistence of alfalfa during its introduction [6]. In this study, the over-wintering rate of alfalfa gradually decreased with the increase in fall-dormancy level. In late autumn and early winter, alfalfa enters the stage of low-temperature domestication and over-wintering hardiness adaptation. Alfalfa varieties have different responses to cold signals and photoperiods at different fall-dormancy levels, resulting in different over-winter survival rates. This is mainly determined by the genetic characteristics of alfalfa, where low fall-dormancy alfalfa store more nutrients in the fall to maintain the necessary nutrient requirements for the winter. The accumulation of soluble sugars plays an important role in adapting to low temperatures [34]. There was a tendency for the soluble sugar content of aboveground stems and leaves to decrease with time under late autumn conditions, mainly due to the transfer of soluble sugars to the underground root system. As fall-dormancy levels decreased, this pattern increased [35,36]. In addition, the enhancement of fall-dormancy in alfalfa increased the antioxidant substances in plant, the soluble substance content in alfalfa roots, and the unsaturated fatty acid content in cell membranes, leading to the increased tolerance to freezing and cold [37]. Compared with previous studies, we found that the over-summering rate of alfalfa was lower than the over-wintering rate, but this difference

was not significant [38], which was mainly influenced by the difference in rainfall during alfalfa growth. Xinjiang is arid and has little rainfall, and alfalfa relies heavily on irrigation for water. At the same time, the rainy season in some parts of mainland China is often accompanied by continuous rainy days, resulting in alfalfa being prone to root rot after mowing and unable to continue growth.

## 5. Conclusions

Our study showed that alfalfa with fall dormancy levels 3–5 was more conducive to the accumulation of C, but the concentrations of N and P were lower in the Shihezi region of Xinjiang. The growth of non-fall-dormant and fall-dormant alfalfa is mainly limited by N, while the growth of medium fall-dormant alfalfa is affected by N and P together. Mature leaves with lower nutrient concentration have greater 'source' ability in nutrient redistribution and reuse during the senescence stage. Different fall-dormancy alfalfa presents different productivity, leaf nutrient concentration, and reabsorption laws. In addition to soil nutrient testing, future fertilization management should also take into account the growth requirements of alfalfa with different fall-dormancy levels or determine the amount of fertilizer to be applied according to the yield potential of the alfalfa.

**Author Contributions:** Conceptualization, Y.S., X.W., C.M. and Q.Z.; methodology, Y.S.; software, Y.S.; validation, Y.S. and Q.Z.; formal analysis, Y.S.; investigation, Y.S.; resources, Q.Z.; data curation, Y.S.; writing—original draft preparation, Y.S.; writing—review and editing, X.W., C.M. and Q.Z.; visualization, Y.S.; supervision, Q.Z.; project administration, Q.Z.; funding acquisition, Q.Z. All authors have read and agreed to the published version of the manuscript.

**Funding:** This research was funded by the National Natural Science Foundation of China (Grant no. 32001400, 32260347), the Fok Ying Tung Education Foundation of China (Grant no. 171099), the Science and Technology Innovation Key Talent Project of Xinjiang Production and Construction Corps (2021CB034), and the China Agriculture Research System of MOF and MARA.

**Institutional Review Board Statement:** Not applicable.

**Informed Consent Statement:** Not applicable.

**Data Availability Statement:** Not applicable.

**Conflicts of Interest:** The authors declare no conflict of interest. The funders had no role in the design of the study; in the collection, analyses, or interpretation of data; in the writing of the manuscript; or in the decision to publish the results.

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
