# Peer review of "Characteristics of Carbon, Nitrogen and Phosphorus Stoichiometry and Nutrient Reabsorption in Alfalfa Leaves with Different Fall-Dormancy Levels in Northern Xinjiang, China"

_agriculture, doi:10.3390/agriculture12122154_

Round 1
Reviewer 1 Report
Revision required

Author Response
Response to Reviewer 1 Comments
Point 1: Title: The article is not focused because it is too long, up to 19 pages, it is better to split it into 2 articles
Response 1: According to the reviewer's opinion, we have partially revised the summary and Conclusion of the article to make it more focused.
Point 2: Abstract: Describe the objective of the research
Response 2: According to the reviewer's opinion, we supplemented the research objective in the abstract. The details are as follows:
“Abstract: Alfalfa productivity and cold resistance in different regions are influenced by the fall-dormancy level of alfalfa. However, it is unclear whether the stoichiometric characteristics and nutrient resorption efficiency in alfalfa leaves also vary with the fall-dormancy level. In order to further understand the differences in nutrient absorption and requirements of different fall-dormant alfalfa, we conducted field trials on 30 different fall-dormancy alfalfa cultivars for two consecutive years in 2020 and 2021. We investigated the concentrations of carbon, nitrogen, and phosphorus in mature and senescent alfalfa leaves, nutrient stoichiometry ratios, and the coupling relationship between nutrient reabsorption efficiency and dry matter yield. The differences in nutrient reabsorption, fall dormancy and dry matter yield of different fall-dormant alfalfa, and the correlation between indicators were utilized to further analyze the regulatory mechanisms of nutrient reabsorption in different fall-dormancy alfalfa varieties. The results demonstrated that the nitrogen reabsorption efficiency (NRE) and phosphorus reabsorption efficiency (PRE) of leaves increased first and then decreased with the increase of fall-dormancy, whereas the carbon reabsorption efficiency (CRE) showed the reverse tendency. Different fall-dormancy alfalfa varieties significantly affected the dry matter yield and nutrient absorption in the first cut, while the last cut had the lowest variable coefficient and impact. There was a significant decrease in the over-winter survival rate of alfalfa as the fall-dormancy level increased, whereas the over-summer survival rate was less affected by the fall-dormancy level. As the growth year increased, there was a significant decrease in the over-winter survival rate. Among mature leaves, the NRE and PRE showed a significant positive correlation with the C concentration, while they showed a significant negative association with the N and P concentrations. In the same cut, the dry matter yield decreased with the increase of CRE but increased with the increases of NRE and PRE, while there was no significant trend in dry matter yield and nutrient resorption efficiency (NuRE) between different cuts. Taken together, the alfalfa survival rate and dry matter yield were relatively better in the moderate fall-dormancy (fall-dormancy level, FD=4, 5) types and fall-dormancy (FD=3) type, with a corresponding increase in the reabsorption requirements for nitrogen and phosphorus.”
Point 3: Abstract: Research methods and data analysis have not been described
Response 3: According to the reviewer's opinion, we supplemented the research methods and data analysis in the abstract. The details are in question Response 2.
Point 4: Abstract: In the first writing should not be abbreviated
Response 4: We are very sorry for our negligence of not adding the full name when the first abbreviation appeared. and now we have added. The details are in question Response 2.
Point 5: Figure 1: Presenting data like this makes it difficult to know exactly how many numbers there are
Response 5: According to the reviewer's opinion, we have redrawn Figure 1 and added the annual accumulated rainfall and annual average temperature. The details are as Figure 1.
Figure 1. Climate data during the experimental period
Point 6: Figure 3: Is this not significant different?
Response 6: I forgot to add “SD*10 means that the standard error should be multiplied by 10” in the caption, and now I have added. The details are as Figure 3.
Figure 3. Stoichiometric ratios of elements in mature alfalfa leaves with different fall-dormancy levels in 2020. SD*10 means that the standard error should be multiplied by 10.
Point 7: Figure 5: Presenting data like this makes it difficult to know exactly how many numbers there are
Response 7: The data cannot be presented well, possibly due to the limitations of the coordinate axis and the color matching of the figure. When drawing the figure, we chose the same coordinate axis to better compare the difference between the year and the cut. In order to compare the data in the figure more clearly, we use light colors to replace the previous colors. The details are as Figure 5.
Figure 5. Stoichiometric characteristics of senescent alfalfa leaves with different fall-dormancy levels.
Point 8: Figure 9: It's hard to immediately understand this image, it's not interactive
Response 8: According to the reviewer's opinion, we adjusted the size of the coordinate axis, text and legend of the picture for a clearer view. The details are as Figure 9.
Figure 9. Relationship among C, N and P, element reabsorption, and dry matter yield in mature leaves.
Point 9: Conclusion: Basically, this research is good, but it is too long, up to 19 pages and not focused, so it is recommended to split it into 2 articles.
Response 9: According to the reviewer's opinion, we have partially revised the summary and Conclusion of the article to make it more focused.

Reviewer 2 Report
This is an interesting and original paper. The authors characterise the carbon, nitrogen, and phosphorus concentrations, as well as the stoichiometry ratio, in alfalfa leaves with varying levels of fall dormancy in Northern Xinjiang, China. The experiment is well-planned and very well-written. Certain minor revisions need to be carried out, given as comments in the attached file. Grammatical errors or incomplete sentences have been underlined in the file, it should be improved. In the discussion section- line number 243-246 looked incomplete, please check. Figures are well prepared, but colour combinations were worse; please remove the blue colour from all graphs, and if possible, use light colours in place of red and blue.

Author Response
Response to Reviewer 2 Comments
Point 1: Grammatical errors or incomplete sentences have been underlined in the file, it should be improved.
Response 1: According to the reviewer's opinion, we have revised the grammatical errors or incomplete sentences.
Point 2: In the discussion section- line number 243-246 looked incomplete, please check.
Response 2: According to the reviewer's opinion, we have revised the incomplete sentences. The details are as follows:
“C, N and P are essential nutrients for plant growth and development, and their stoichiometric characteristics can reflect the intra-plant stoichiometric endostability and elemental interrelationships [3]. This study showed that alfalfa fall-dormancy levels could significantly affect leaf C, N and P concentrations, with the medium fall-dormancy type (level 4) alfalfa obtaining the highest C and the lowest N and P concentrations. Based on previous studies, we think the concentrations of C, N, and P in different fall-dormancy alfalfa are inconsistent because of the following two aspects. On the one hand, the response of different fall-dormant alfalfa to the external environment is inconsistent, with the most significant effect of soil temperature and light duration, and the effect of light is about 75% of the soil temperature. The effect of soil temperature in the early growth period of medium fall-dormancy alfalfa is close to that of fall-dormancy alfalfa, while the effect of light duration is only 1/2 of that of fall-dormancy alfalfa varieties. However, as time goes on, the effect of light on moderately fall-dormancy alfalfa varieties decreased significantly, while the non-fall-dormancy alfalfa varieties were influenced only by soil temperature and no longer by light. This has an effect on the whole-plant biomass, photosynthetic carbon fixation, root nutrient uptake and persistence after alfalfa re-greening [20]. On the other hand, because the low fall-dormancy alfalfa enters a dormant state very early in the previous fall, it consumes less nutrients in the lower ground root system and stores a large amount of carbohydrates and proteins for alfalfa growth after overwintering and re-greening [21]. In this study, we also found that alfalfa dry matter yield was highest at fall-dormancy levels 4, 3, and 5, and that the higher yield increased nutrient demand, leading to a decrease in soil nutrient content, which reduced the nutrient cycling and metabolic activity, ultimately leading to a reduction in the nutrient feedback mechanism between alfalfa and soil. While higher carbon fixation also diluted the N and P content in plant, and the low biomass of alfalfa with high and low fall-dormancy levels might cause a nutrient "concentration" effect [18].
The interrelationship between the major nutrients and their stoichiometric ratios in plant is a combination of the plant's nutrient allocation strategy and adaptation to the external environment. A high C: N and C: P ratios indicate that plants use N and P more efficiently, which is a strategy for adapting to soil nutrient depletion during growth [22]. In this study, medium fall-dormancy alfalfa had higher carbon-to-nitrogen and carbon-to-phosphorus ratios, mainly due to the soil depletion caused by the high yields for three to four consecutive years. Nitrogen-phosphorus ratio can reflect the dynamic balance between soil nutrient content and plant nutrient demand to a certain extent. It is possible to identify the main limiting nutrients for alfalfa growth by analyzing its nitrogen-phosphorus concentration and nitrogen-phosphorus ratio. Koerselman [23] suggested that alfalfa growth is mainly limited by nitrogen when the nitrogen-phosphorus ratio is less than 14, and when the nitrogen-phosphorus ratio is greater than 16, the alfalfa growth is mainly limited by phosphorus. Alfalfa growth was limited by a combination of nitrogen and phosphorus when the nitrogen-phosphorus ratio was between 14 and 16. In this study, for the most of the fall-dormancy and non-fall-dormancy alfalfa types, the nitrogen-phosphorus ratios were less than 14, and they were mainly limited by N. Most of the medium fall-dormancy types were between 14 and 16 and were limited by a combination of N and P. This may be mainly due to the fact that the medium fall-dormancy alfalfa in Shihezi region is more suitable for growth and requires more soil nutrients to be consumed during growth, so this type is more nutrient-limited. However, some studies suggest that alfalfa growth is more phosphorus deficient. On the one hand, in this study, extra phosphorus fertilizer was added, resulting in sufficient available phosphorus for the normal growth of alfalfa. On the other hand, alfalfa, as a legume, can use rhizobia to convert the nitrogen in air into available nitrogen. Some fertilization studies have shown that moderate additions of nitrogen fertilizer can significantly increase alfalfa yield [24], indicating to some extent that the nitrogen fixed by alfalfa rhizobia is not sufficient to meet the growth requirements of alfalfa. In addition, rhizobia in soil have limitations in fixing atmospheric N. The higher yield of moderate fall-dormancy alfalfa does not match the corresponding increase in N fixation by rhizobia. Not only that, at higher soil water holding capacity, the N: P in alfalfa leaf also tends to decrease gradually with increasing growth time [25].
”
Point 3: Figures are well prepared, but colour combinations were worse; please remove the blue colour from all graphs, and if possible, use light colours in place of red and blue.
Response 3: According to the reviewer's opinion, we have used light colours in place of red and blue.
Point 4: Keywords: Please rearrange in alphabetically
Response 4: According to the reviewer's opinion, we have Arranged the keywords in alphabetic order. The details are as follows:
“Keywords: alfalfa; carbon, nitrogen and phosphorus stoichiometry; fall-dormancy; nutrient reabsorption”

Reviewer 3 Report
The manuscript entitled (Characteristics of carbon, nitrogen and phosphorus stoichiometry and nutrient reabsorption in alfalfa leaves with different fall-dormancy levels in Northern Xinjiang, China) is well written and minor comments should be taken into consideration before acceptance.
Arrange the keywords in alphabetic order.
The abbreviations in the abstract such as CRE, NRE, and PRE should be mentioned first in the full name.
Line 99, Gossypium spp. should be in italic.
The aim needs to be well presented.
Add more recent references in many parts.
The conclusion needs to be improved and the significance of the work should be well presented.
Author Response
Response to Reviewer 3 Comments
Point 1: Arrange the keywords in alphabetic order.
Response 1: According to the reviewer's opinion, we have Arranged the keywords in alphabetic order. The details are as follows:
“Keywords: alfalfa; carbon, nitrogen and phosphorus stoichiometry; fall-dormancy; nutrient reabsorption”
Point 2: The abbreviations in the abstract such as CRE, NRE, and PRE should be mentioned first in the full name.
Response 2: We are very sorry for our negligence of not adding the full name when the first abbreviation appeared. and now we have added. The details are as follows:
“Abstract: Alfalfa productivity and cold resistance in different regions are influenced by the fall-dormancy level of alfalfa. However, it is unclear whether the stoichiometric characteristics and nutrient resorption efficiency in alfalfa leaves also vary with the fall-dormancy level. In order to further understand the differences in nutrient absorption and requirements of different fall-dormant alfalfa, we conducted field trials on 30 different fall-dormancy alfalfa cultivars for two consecutive years in 2020 and 2021. We investigated the concentrations of carbon, nitrogen, and phosphorus in mature and senescent alfalfa leaves, nutrient stoichiometry ratios, and the coupling relationship between nutrient reabsorption efficiency and dry matter yield. The differences in nutrient reabsorption, fall dormancy and dry matter yield of different fall-dormant alfalfa, and the correlation between indicators were utilized to further analyze the regulatory mechanisms of nutrient reabsorption in different fall-dormancy alfalfa varieties. The results demonstrated that the nitrogen reabsorption efficiency (NRE) and phosphorus reabsorption efficiency (PRE) of leaves increased first and then decreased with the increase of fall-dormancy, whereas the carbon reabsorption efficiency (CRE) showed the reverse tendency. Different fall-dormancy alfalfa varieties significantly affected the dry matter yield and nutrient absorption in the first cut, while the last cut had the lowest variable coefficient and impact. There was a significant decrease in the over-winter survival rate of alfalfa as the fall-dormancy level increased, whereas the over-summer survival rate was less affected by the fall-dormancy level. As the growth year increased, there was a significant decrease in the over-winter survival rate. Among mature leaves, the NRE and PRE showed a significant positive correlation with the C concentration, while they showed a significant negative association with the N and P concentrations. In the same cut, the dry matter yield decreased with the increase of CRE but increased with the increases of NRE and PRE, while there was no significant trend in dry matter yield and nutrient resorption efficiency (NuRE) between different cuts. Taken together, the alfalfa survival rate and dry matter yield were relatively better in the moderate fall-dormancy (fall-dormancy level, FD=4, 5) types and fall-dormancy (FD=3) type, with a corresponding increase in the reabsorption requirements for nitrogen and phosphorus.”
Point 3: Line 99, Gossypium spp. should be in italic.
Response 3: According to the reviewer's opinion, we have changed the word " Gossypium spp." to italic. The details are as follows:
“Alfalfa was planted on April 27, 2018, by manual bunch planting, and to avoid the autotoxic allelopathy caused by continuous cropping, the fields with many years of other crop rotations were selected as test plots, where the previous crop was cotton (Gossypium spp.).”
Point 4: The aim needs to be well presented.
Response 4: According to the reviewer's opinion, we supplemented the research objective in the abstract. The details are in question Response 2.
Point 5: Add more recent references in many parts.
Response 5: According to the reviewer's opinion, we have replaced the old references with the latest references.
Point 6: The conclusion needs to be improved and the significance of the work should be well presented.
Response 6: According to the reviewer's opinion, we have partially modified and supplemented the conclusions. The details are as follows:
“Our study in the Shihezi region of Xinjiang showed that C in mature and senescent alfalfa leaves tended to increase first and then decrease, whereas N and P both tended to decrease first and then increase as the fall-dormancy levels increased. Nutrient reabsorption by alfalfa decreased as the cut decreased, with medium-fall dormancy alfalfa reabsorbing more N and P, but the increase was not significant in the final cut. The yield of middle fall dormancy type and fall dormancy type alfalfa was significantly higher than that of other fall dormancy type alfalfa, and alfalfa yield was positively correlated with NRE and PRE, and negatively correlated with N and P in green leaves. Alfalfa persistence showed a decreasing trend with the increase of fall-dormancy levels, and the differences in yield and persistence caused by different fall-dormancy levels must be considered in the selection of alfalfa varieties. The decreasing trend was more obvious in 2021 than in 2020, which may be caused by the growth year. In addition to soil testing and fertilization, future fertilization management should also take into account the growth requirements of alfalfa with different fall-dormancy levels or determine the amount of fertilizer to be applied according to the yield of alfalfa.”
